# Synthetic Neuraminidase Vaccine Induces Cross-Species and Multi-Subtype Protection

**DOI:** 10.3390/vaccines13040364

**Published:** 2025-03-28

**Authors:** Matthew J. Pekarek, Erika M. Petro-Turnquist, Nicholas E. Jeanjaquet, Kristine V. Hoagstrom, Enzo LaMontia-Hankin, Leigh Jahnke, Adthakorn Madapong, Eric A. Weaver

**Affiliations:** 1School of Biological Sciences, University of Nebraska-Lincoln, Lincoln, NE 68583, USA; 2Nebraska Center for Virology, University of Nebraska-Lincoln, Lincoln, NE 68583, USA; 3Department of Chemistry and Biochemistry, Nebraska Wesleyan University, Lincoln, NE 68504, USA; 4College of Arts and Sciences, University of Nebraska-Lincoln, Lincoln, NE 68583, USA

**Keywords:** influenza A virus, vaccine, neuraminidase, H1N1, H5N1, centralized, consensus, adenovirus, viral vector

## Abstract

The genetic diversity of influenza A virus is a major obstacle that makes vaccine effectiveness variable and unpredictable. **Objectives**: Current vaccines induce strain-specific immunity that oftentimes fail to protect against divergent strains. Our previous research explored synthetic centralized consensus (CC) vaccines to minimize immunogen-strain divergence and focused on the viral glycoprotein hemagglutinin. **Methods**: Recently, emerging evidence of neuraminidase (NA)-mediated immunity has shifted vaccine strategies, prompting our development of a CC NA type 1 (N1CC) vaccine based on ancestral N1 sequences and delivered using a human adenovirus type 5 vector **Results**: The N1CC vaccine elicited antibody responses with NA inhibition activity and induced NA-specific T-cell responses. In lethal influenza challenge models, N1CC fully protected mice from death against human, swine, and avian influenza H1N1 and H5N1 strains. **Conclusions**: These findings support NA as a protective immunogen and demonstrate the power and efficacy of a centralized consensus NA design.

## 1. Introduction

Influenza A viruses (IAVs) are a primary cause of respiratory infections in humans. Infections can result in a range of mild symptoms, such as a cough and sore throat, to more severe outcomes like pneumonia and death [1]. Yearly epidemics and global spread of IAVs occur despite our knowledge of IAV circulation [2]. The genetic variation of IAV is the driving cause of this sustained spread. IAVs are classified into subtypes based on their hemagglutinin (HA) and neuraminidase (NA) glycoproteins. Currently, there are 18 HA subtypes and 11 NA subtypes that have been characterized. Further, a majority of IAV subtypes only circulate in non-human species, such as birds, swine, and other mammals [1,3,4]. Spillover events from one host species to another have been well documented [5,6,7]. These spillover events often lead to severe infection or death in the new host or trigger pandemic outbreaks of influenza when transmitted to humans [8,9]. Therefore, improved prevention strategies against IAV must consider both human and non-human IAV genetic variation.

Vaccination is the primary method to prevent influenza infection and disease. However, current vaccination methods tend to induce strain-specific responses [10]. Current vaccines are typically comprised of two IAV strains and one or two influenza B virus (IBV) strains [11] and commonly target the HA glycoprotein [12,13]. The effectiveness of these vaccines often varies due to numerous factors, including virus subtype, study population and location, and influenza season [11,14,15,16]. One method to address this variability is through consensus immunogen design. Consensus immunogens attempt to minimize the genetic distance of a random wild-type sequence from the vaccine immunogen [17]. This reduces the probability of circulating strains being mismatched from the vaccine immunogen throughout the flu season. Several different consensus HA vaccines have been reviewed previously [13]. However, a major limitation of consensus vaccine designs is the inherent bias towards protection against clusters of strains with high sequence prevalence due to overrepresentation in the input sequence population. Our previous work has overcome this by utilizing a centralized consensus design. Centralized consensus immunogens still minimize the genetic distance from natural sequences but limit sequencing biases by selecting representative strains from major branches for immunogen design. Centralized consensus vaccines have previously shown promising results when tested against H1 [17,18,19], H3 and H5 [18,19], and H2 [20] IAVs. Another recently explored strategy for universal flu vaccine development involves targeting NA [21]. Previous work has shown the potential for NA vaccines to contribute to protection through vaccination with or without HA included [22,23,24]. More recently, reports that targeted NA across a variety of influenza types, vaccine immunogen designs, vaccine delivery platforms, and animal models further support the potential of NA as a protective component for influenza vaccines [25,26,27,28,29]. These studies have identified both antibody production and cell-mediated immune activation that can lead to protection against infection. Further, previous consensus NA designs have shown promising signs of protective efficacy in the mouse model [30,31]. When analyzing all these results from various models, NA-based vaccines point to promising protection against influenza infection, either alongside or independent of HA.

Our previous work has detailed the protective efficacy of centralized consensus HA vaccines [17,18,19,20] However, the ability of a centralized consensus NA immunogen to elicit the same protection as these HA vaccines was unclear. Thus, we designed a centralized consensus N1 immunogen to investigate its protective efficacy. We based the N1 centralized consensus sequence (N1CC) on a representative population of clinically relevant human N1 sequences isolated before 1990. The N1CC sequence was designed prior to the emergence of pandemic 2009 lineage H1N1; therefore, these sequences were more representative at the time of design. Our results show that vaccination with a human adenovirus type 5 (HAdV-5)-vectored N1CC vaccine induced detectable antibody and T-cell responses against multiple HxN1 influenza strains. Further, a single vaccination protected mice from challenge with highly genetically diverse influenza strains isolated from multiple hosts. Finally, N1CC vaccination protected mice from lethal challenge after dose de-escalation with vaccine doses 100–1000 times lower than common pre-clinical influenza vaccine trial doses [18,32,33]. Together, our results provide the first evidence of a centralized consensus NA vaccine providing protection in the mouse model and further support for continued investigation of centralized consensus NA immunogens as broadly protective influenza vaccine candidates.

## 2. Materials and Methods

### 2.1. Animals and Research Ethics

Six- to eight-week-old female BALB/c mice Jackson Laboratory (Bar Harbor, ME, USA) were maintained under the Association for Assessment and Accreditation of Laboratory Animal Care (AAALAC) guidelines at the University of Nebraska—Lincoln (UNL). All experiments in this study were approved under the UNL Institutional Animal Care and Use Committee (IACUC) under protocol number 2662 (approved on 30 September 2024) and the UNL Institutional Biosafety Committee (IBC) under protocol 619. Animal experiments were performed according to the guidelines set forth by the Animal Welfare Act, the Animal Welfare Policy, and the NIH Guide for the Care and Use of Laboratory Animals.

### 2.2. Influenza Viruses

A/Puerto Rico/8/1934 (PR8, PR/34) (VR95) was obtained from American Type Culture Collection (Manassas, VA, USA). A/Fort Monmouth/1/1947 (FM47) (NR-2759) and A/swine/1976/1931 (swUSA31) (NR-3168) were obtained from the Biodefense and Emerging Infectious Disease Repository (Manassas, VA, USA). A/swine/Minnesota/A01489606/2015 (swMN15) (A01489606) was obtained from the United States Department of Agriculture swine influenza repository (Ames, IA, USA). Recombinant (6+2 A/PR/8/34) A/Vietnam/1203/2004 (Viet04) and A/bar-headed goose/Qinghai/A/2005 (Goose05) without the multi-basic cleavage site in the HA were obtained from Dr. Richard Webby at St. Jude Children’s Research Hospital (Memphis, TN, USA). Mouse-adapted strains from each stock were produced as previously described [34] for all viruses except swUSA31, which was obtained from BEI (NR-31681). All mouse-adapted viruses were serially passaged in mice between 4 and 12 times in total. Virus stocks were amplified in 10- to 11-day-old specific pathogen-free embryonated chicken eggs (Charles River Laboratories, Wilmington, MA, USA) and incubated at 35 °C for 48–72 h. Allantoic fluid was then harvested from infected eggs, with virus titers quantified through hemagglutination units, and aliquots were stored at −80 °C. Mouse-adapted strains were passaged through embryonated eggs only once before mouse challenge. Median lethal doses were titered in 8–12-week-old BALB/c mice.

### 2.3. Immunogen Design and Vaccine Production

The centralized consensus ancestral N1 (N1CC) was designed by creating a consensus sequence based on representative ancestral N1 sequences available on Genbank. The accession numbers for the sequences used for the immunogen design are as follows: CY020287.1, CY009278.1, AF250356.2, CY009598.1, CY008990.1, CY147376.1, CY009614.1, CY009334.1, CY009326.1, L25815.1, J02146.1, CY020479.1, CY121880.1, and J02177.1 (Table 1). The sequences were downloaded from Genbank and aligned by ClustalW using Geneious (v 11.1.5). Any position in the consensus sequence identified as an ambiguous residue was replaced manually with one of the most common residues in the representative population. The centralized consensus H1 (H1CC) and H5 (H5CC) immunogens were designed similarly to the N1CC as previously described [17,18]. The N1CC gene was synthesized by Genscript, Inc. (Piscataway, NJ, USA) and further cloned into a replication-defective human adenovirus subtype 5 (HAdV-5) vector using the Ad-Easy Adenoviral Vector System (Agilent, Santa Clara, CA, USA) as previously described [35]. Restriction enzymes used for cloning were purchased through New England Biolabs (Ipswich, MA, USA) and transfection reagents were purchased through Qiagen (Germantown, MD, USA). Amplification and propagation steps were conducted using increasing surface areas of HEK293 cells before a final collection of virus particles from a 10-cell stack (Corning) was complete. N1CC vaccine was purified through two CsCl gradient centrifugation purification steps and desalted with Econo-Pac 10DG Desalting Columns (Bio-Rad, Hercules, CA, USA) before being diluted in a 1:2 ratio with 10% glycerol and stored at −80 °C. Total viral particles were quantified based on OD260 readings on a NanoDrop™ Lite Spectrophotometer (Thermo Fisher Scientific, Waltham, MA, USA), and the infectious unit-to-virus particle ratio was calculated using the AdenoX Rapid Titer kit (Clontech Laboratories, Mountain View, CA, USA). Production of H1CC and H5CC vaccines has previously been described [17,18].

### 2.4. Phylogenetic Analysis

Phylogenetic analysis was conducted using Geneious v11.0.5. Unique N1 sequences isolated from humans, avian species, and swine up to 2007 available on the Influenza Research Database as of 26 October 2022 were downloaded using the following search criteria: complete NA sequence, N1 subtype, isolated up to 2007, exclude duplicate sequences, exclude lab strains, host species avian/human/swine. Once downloaded, sequences with duplicate strain names were removed. Sequence alignments were done using the ClustalW plug-in on Geneious using the following parameters: BLOSUM cost matrix, gap open cost = 10, gap extend cost = 0.1. Phylogenetic trees created in Geneious were created using the following parameters: Jukes–Cantor distance model, neighbor-joining method, no outgroup.

### 2.5. Western Blot

Cellular expression of NA was probed by western blot of N1CC-infected HEK293 cells. Confluent 12-well plates were infected with N1CC at a multiplicity of infection of 50 infectious units per cell and incubated at 37 °C with 5% CO_2_. After 48 h, cells were harvested, and proteins were denatured by resuspending the cells in Laemmli buffer with 2-mercaptoethanol and heating at 100 °C for 10 min. Cell debris was removed using a QIAshredder (Qiagen, Germantown, MD, USA). Denatured protein samples were resolved using a 10% SDS-PAGE and transferred to a nitrocellulose membrane. The membrane was blocked using 5% dry non-fat milk in TBST for 90 min before the addition of the primary goat polyclonal anti-A/New Jersey/8/1976 NA antiserum (BEI Resources, Manassas, VA, USA), NR-3136). The membrane was incubated in primary antiserum at 4 °C overnight with rocking. GAPDH was used as a cellular loading control and was detected using a mouse monoclonal anti-GAPDH antibody conjugated with horseradish peroxidase (HRP) (Santa Cruz Biotechnologies (Dallas, TX, USA), #0411). For NA detection, the membrane was washed three times to remove excess primary antiserum and incubated with donkey anti-goat IgG conjugated with HRP (R&D Systems, HAF109). After washing three times with TBST, the membrane was developed using the SuperSignal West Pico Chemiluminescent Substrate (Thermo Scientific, Waltham, MA, USA) for five minutes before imaging. All antibodies were diluted in 1% milk with TBST.

### 2.6. Mouse Vaccinations and Sample Collections

For serum and splenocyte analysis, six- to eight-week-old female BALB/c mice (n = 10) were vaccinated intramuscularly with 10^9^ virus particles of N1CC, H1CC, or H5CC in a 50 µL dose split between both quadriceps. Vaccinated mice were compared to mice with a mock vaccine delivering 50 µL of DPBS. Vaccination was performed while under isoflurane anesthesia. Three weeks post-vaccination, half (n = 5) of the mice were boosted with a homologous dose of N1CC, H1CC, H5CC, or DPBS, again delivered intramuscularly. The remaining mice (n = 5) from each group were sacrificed for serum and splenocyte isolation. Serum was collected via cardiac puncture, separated from whole blood using BD Microtainer^®^ Chemistry Tubes (Becton-Dickinson, Franklin Lakes, NJ, USA), and stored at −80 °C. Splenocytes were processed through a 40 µm nylon mesh cell strainer (FisherBrand, Waltham, MA, USA) to create single-cell suspensions, and red blood cells were lysed using ammonium-chloride-potassium (ACK) lysis buffer. Processed splenocytes were suspended in 90% *v*/*v* fetal bovine serum (FBS) and 10% *v*/*v* DMSO (bio-world, Irving, TX, USA) and stored in a liquid nitrogen vapor phase. Two weeks post-boost, the remaining five mice were sacrificed for serum and splenocyte isolation as described above.

### 2.7. Influenza Virus Microneutralization

Serum from vaccinated mice was heat inactivated at 56 °C for 30 min then cooled at 4 °C. In a sterile, 96-well plate, serum was initially diluted 1:10 in DMEM (Cytiva, Marlborough, MA, USA) supplemented with 5% FBS. Then, serum was serially diluted twofold, and 100 TCID_50_ units of virus were added. The serum-virus mixture was incubated at 37 °C with 5% CO_2_ for one hour. Following incubation, 2 × 10^4^ Madin–Darby canine kidney (MDCK) cells were added to each well, and plates were incubated at 37 °C with 5% CO_2_ for 24 h. After 24 h, the plates were washed with sterile DPBS before adding DMEM supplemented with 2 µg/mL TPCK-treated trypsin and stored at 37 °C with 5% CO_2_ for 72 h. To read the results, 50 µL 0.5% rooster red blood cells were added to the wells, and agglutination patterns were read after 30- to 45-min incubations at room temperature.

### 2.8. Enzyme-Linked Immunosorbent Assay (ELISA)

High protein-binding plates (Immulon 4 HBX, VWR) were coated overnight at 4 °C with 150 ng per well of recombinant NA protein from A/Puerto Rico/8/1934 (NR-19235, BEI Resources, Manassas, VA, USA) or A/New Caledonia/20/1999 (NR-43779, BEI Resources, Manassas, VA, USA) diluted in carbonate/bicarbonate buffer. Protein was removed, and plates were washed four times with DPBS with 0.05% Tween-20 (DPBS-T) before blocking with 10% skim milk in DPBS-T for two hours at room temperature. Serum samples were heat-inactivated at 56 °C for 45 min before diluting 1:100 in 5% skim milk. Plates were washed three times with DPBS-T, and diluted serum in duplicate was incubated on the coated plates for one hour at room temperature. Excess serum was removed, and wells were washed 5 times with DPBS-T before addition of goat anti-mouse IgG-HRP conjugate (AP308P, Millipore Sigma, Burlington, MA, USA) and incubated for 30 min. After removing any excess secondary antibody, 1-Step™ Ultra TMB-ELISA (Thermo Scientific, Waltham, MA, USA) was added to each well and incubated for 15 min, and the reaction was ended upon addition of 2M sulfuric acid. Absorbance was read on a SpectraMax i3x (Molecular Devices, San Jose, CA, USA) plate reader at 450 nm wavelength, and background absorbance without serum added was subtracted from sample OD_450_.

### 2.9. Enzyme-Linked Lectin Assay (ELLA)

The ELLA procedure was adapted from [36]. Serum samples were heat-inactivated at 56 °C for 45 min, diluted to an initial 1:10 dilution, and further twofold serially diluted in a 96-well plate. A plate coated with fetuin (Sigma-Aldrich, Saint Louis, MO, USA) for at least 18 h was washed three times with DPBS-T. An amount of 50 µL of each serum dilution was transferred onto the fetuin-coated plate before adding 50 µL of diluted wild-type replication-competent influenza virus. Optimal virus concentration was determined as the lowest virus dilution to reach an OD_450_ ≥ 75% maximum undiluted virus readout while maintaining a ≤10% OD_450_ no-virus control background absorbance. Serum-virus mixtures were incubated for 18 h at 37 °C with 5% CO_2_. After incubation, the plates were washed six times with DPBS-T before adding 100 µL of peanut agglutinin-horseradish peroxidase conjugate (PNA-HRPO, Sigma-Aldrich, Saint Louis, MO, USA) to all wells and incubated at room temperature for two hours. Plates were washed three times with DPBS-T before 100 µL 1-Step™ Ultra TMB-ELISA (Thermo Scientific) equilibrated to room temperature were added to each well and developed for 15 min. Development was stopped by the addition of 100 µL 2M sulfuric acid. Absorbance was read on a SpectraMax i3x (Molecular Devices) plate reader at 450 nm wavelength. Serum samples were run in duplicate. Endpoint NA inhibitory titers were determined based on the average absorbance of the duplicate wells after normalizing to background controls where ≥50% inhibition of OD_450_ was observed.

### 2.10. Enzyme-Linked Immunosorbent Spot (ELISpot)

T-cell responses to vaccination were measured using IFN-γ ELISpot assays. Overlapping peptide libraries for PR8 HA (NR-18973) and NA (NR-19257), A/Thailand/4(SP-528)/2004 (Thai04) HA (NR-2604), and Viet04 NA (NR-19258) were provided by the BEI resource repository. Peptides from each individual library were pooled to a final concentration of 5 µg/mL. Several 96-well hydrophobic high protein-binding plates containing Immobilon-P Membrane (Millipore) were coated with 100 ng anti-mouse IFN-γ monoclonal antibody AN18 (Mabtech) overnight. Coated plates were then blocked with RPMI-1640 supplemented with 5% *v*/*v* fetal bovine serum and 1% penicillin/streptomycin at 37 °C with 5% CO_2_ for two hours. Fifty µL of pooled peptide were added to experimental wells for restimulation of vaccinated splenocytes, with 50 µL concanavalin A (5 µg/mL) (Sigma-Aldrich, Saint Louis, MO, USA) added to positive control wells and 50 µL RPMI-1640 with 5% fetal bovine serum and 1% penicillin/streptomycin added to unstimulated control wells. Splenocytes at a concentration of 2.5 × 10^5^ cells per well were added to the plate in duplicate and stored at 37 °C with 5% CO_2_ for 18 h. Plates were then washed six times with DPBS-T before adding 50 µL biotinylated anti-mouse IFN-γ monoclonal antibody R4-6A2 (Mabtech, Cincinnati, OH, USA) at 1 µg/mL. Plates were left for one hour at room temperature before six washes with DPBS-T. Following the wash steps, 100 µL of diluted streptavidin-alkaline phosphatase (Mabtech) were added to all wells. Plates were incubated at room temperature for 45 min and again washed six times with DPBS-T. To develop the plates, 100 µL BCIP/NBT (MOSS Bio., Hanover, MD, USA) alkaline phosphatase substrate were added, and the plates were incubated at room temperature until visible spots appeared in concanavalin A control wells. Plates were then washed repeatedly with deionized water to end the spot formation. Developed plates were dried and stored in the dark for at least 18 h. Spots were counted using an automatic count function ELISpot plate reader (CTL Cellular Technology, Shaker Heights, OH, USA), and data are reported in spot-forming units per million splenocytes (SFU/10^6^ splenocytes).

### 2.11. Mouse Challenge Studies

For vaccine comparison challenges, groups of female BALB/c mice (n = 5) were vaccinated with 10^9^ virus particles of N1CC, H1CC, or H5CC or 50 µL DPBS for mock vaccination. The vaccine dose was previously determined to provide optimal levels of protection [37]. Three weeks post-vaccination, mice were infected with 100 times the median lethal dose (100 MLD_50_) for all three challenge strains (PR8, swUSA31, or Viet04). The virus was delivered intranasally under ketamine/xylazine anesthesia in a 20 µL dose. Mice were then monitored for 14 days post-infection for morbidity and mortality. Twenty-five percent body weight loss compared to weight at virus challenge was used as a human threshold for euthanasia. Three weeks post-vaccination, mice were challenged with 100 MLD_50_ of the challenge strain except for swMN15, which was 20 MLD_50_. Mice were again monitored for 14 days using the same 25% weight loss threshold for humane sacrifice as described above.

### 2.12. Statistical Analysis

All statistical analysis was completed using GraphPad Prism software v.10.0.

## 3. Results

### 3.1. Design of N1 Centralized Consensus Sequence and Construction of Vaccine

To create our centralized consensus N1CC immunogen, we selected a representative sequence population containing 14 unique N1 sequences isolated from 1918–1986 (Table 1). These sequences were chosen based on the year of isolation and clinical relevance to limit bias against the limited older available sequences. The N1CC sequence was created by inserting the most common amino acid at each position along the NA protein. Due to the design taking place prior to the emergence of the pandemic 2009 H1N1 lineage, this limits the bias of a consensus design towards this highly sampled lineage. Further, we hypothesized that this could increase the likelihood of the N1CC immunogen to protect against diverse HxN1 strains due to a wider time period being sampled for consensus design. The phylogenetic relationship of the N1CC sequence with its input population can be seen in Figure 1A. Prior to the emergence of the 2009 pandemic lineage, there were spillover events from avian H5N1 influenza that led to local disease outbreaks and a concern of pandemic spread [38], and these spillover events into mammalian hosts continue to be documented to the present [39]. To visualize the genetic relationship between the N1CC and more divergent N1 sequences isolated from non-human hosts, we created a phylogenetic tree containing all unique N1 sequences isolated from human, swine, and avian species prior to 2008. We observed that the N1CC sequence was more closely related to other huH1N1 sequences than available swine N1 (swH1N1) or avian N1 sequences (Figure 1B). As expected, the N1 genes of the human avian and swine origin viruses cluster together, indicating a shared evolutionary path. The N1CC is based on human-origin strains and is, therefore, localized within the human cluster (Figure 1B). The relative genetic diversity is shown in the phylogenetic tree, and the actual genetic distance is calculated in Table 2.

Due to our previous success in delivering centralized consensus HA vaccines via an adenovirus vector [17,18,19], we chose this as our vector platform for vaccination [40]. The N1CC immunogen sequence was cloned into the E1 region of a ΔE1-ΔE3 deleted HAdV-5 vector genome. Protein expression was confirmed through western blot of infected HEK293 cell lysates using anti-NA polyclonal serum as a primary antibody (Figure 1C).

### 3.2. Functional Antibody Assessment

To characterize antibody responses induced by N1CC vaccination, groups of 10 mice were immunized with 10^9^ virus particles. HAdV-5-vectored H1 centralized consensus (H1CC) or H5 centralized consensus (H5CC) immunogens were included as controls to identify N1CC-specific immune responses, and all HAdV-5-vectored groups were compared to a PBS mock vaccination. Three weeks post-immunization, five mice per group were sacrificed for serum and spleen collection, and the remaining mice were boosted with a homologous vaccine. Boosted mice were sacrificed for serum and spleen collection two weeks post-boost. The percentage of the identity between the strains used in the study and the vaccine immunogens is shown in Table 2.

Microneutralizations were conducted on prime-only and prime-boost serum samples to determine if antibodies induced by vaccination possessed neutralizing activity. Little neutralization was detected after prime-only vaccination for all strains tested except for PR8, where low levels of neutralization were detected from H1CC and N1CC serum (Figure 2A). Only one sample in the N1CC group detected a neutralizing titer > 10. It is possible that vaccination may have led to a unique antibody development in this animal that induced this neutralization. However, this occurrence was likely sample specific since it was observed in only one mouse. The second immunization led to significantly higher neutralization activity detected from the H1CC and H5CC vaccinated serum against homosubtypic strains (huH1N1 and H5N1, respectively) (Figure 2B), but no neutralizing activity was detected after N1CC vaccination.

To assess NA-specific antibodies, we performed an enzyme-linked immunosorbent assay (ELISA) and an enzyme-linked lectin assay (ELLA) to determine whether antibodies were induced by N1CC vaccination and if those antibodies could prevent NA activity. Using a 1:100 dilution of serum harvested from prime-boost immunized mice, we detected NA-binding antibodies only in the N1CC vaccinated serum against recombinant NA from NC99 and PR8 (Figure 3A). While we were unable to assess NA-inhibiting antibodies against NC99, we did discover that low levels of detectable NA-inhibiting antibody titers correlated with the percentage of amino acid identity between the strains’ NA sequence and the N1CC sequence (Figure 3B, Table 2). Through these results, we can conclude that N1CC vaccination can induce detectable antibody production, though functionality remains unclear.

### 3.3. Induction of T-Cell Responses

We next quantified antigen-specific T-cell activation after vaccination using IFN-γ ELISpot. After a single immunization, IFN-γ-producing T-cells were detected by restimulating isolated splenocytes with an overlapping peptide library containing the PR8 or Viet04 NA sequence (Figure 4A, left). H1CC and H5CC vaccination induced cross-reactive T-cell responses to overlapping peptide libraries in both the homosubtypic and heterosubtypic HA (Figure 4A, right). This inter-subtype cross-reactivity has been previously described for another computationally designed, viral-vectored vaccine due to cross-reactivity in the C-terminal end of the H1 and H5 HA proteins [41]. No boosting of T-cell responses after N1CC boost immunization was observed (Figure 4B, left) (1207 SFU/10^6^ splenocytes from prime immunized mice after PR8 peptide stimulation, 1304 SFU/10^6^ splenocytes from boost immunized mice after PR8 peptide stimulation), but HA-specific IFN-γ^+^ T-cells were increased after H1CC or H5CC boost immunization (Figure 4B, right). Our experiments did not reveal any cross-reactive T-cell response between the HA and NA proteins using the centralized consensus immunogens.

### 3.4. Protection from HxN1 Challenge

To determine whether N1CC vaccination could protect from influenza infection, we performed challenge studies using mouse-adapted HxN1 influenza strains. Mice were vaccinated once with 10^9^ virus particles of N1CC, H1CC, or H5CC or mock vaccinated with DPBS. Three weeks post-vaccination, mice were challenged with 100 times the median lethal dose of a representative huH1N1, swH1N1, or H5N1 strain of influenza. When challenged with PR8, a closely related huH1N1 strain, N1CC vaccination completely protected mice from both weight loss and mortality after vaccination, mirroring the corresponding H1CC vaccination (Figure 5A,B). However, H5CC vaccination did not lead to protection from PR8 challenge (Figure 5A,B) despite the induction of cross-reactive T-cells after vaccination (Figure 4A, right). After challenge with the swH1N1 strain swUSA31, both N1CC and H1CC vaccination completely protected mice from mortality, with only mild weight loss observed. Once again, H5CC vaccination did not protect the mice, with all mice in the group succumbing to the infection (Figure 5C,D). Despite no detected NA-inhibiting antibody response, N1CC vaccination completely protected mice from lethal Viet04 infection. More weight loss was observed after H5N1 infection than PR8 or swUSA31 infection, but weight loss plateaued by four days post-infection, and the animals started regaining weight by seven days post-infection (Figure 5E,F). H5CC vaccination completely protected the animals from weight loss and mortality from homosubtypic challenge, while H1CC vaccination was unable to protect mice from heterosubtypic weight loss or mortality (Figure 5E,F).

### 3.5. Strength and Breadth of Ad-N1CC Protection

To further support our initial protection data, we expanded our challenge panel to include challenge studies for all strains analyzed in the immune correlate assays. Mice were vaccinated with a single dose of 10^9^ vp/mouse intramuscularly. Three weeks post-vaccination, the mice were challenged with three additional lethal strains of influenza virus from human, swine, and avian origins, A/Fort Monmouth/1/1947 (H1N1), A/swine/USA/1976/1931 (H1N1), and A/bar-headed goose/Qinghai/1A/2005 (H5N1), respectively. Based on weight loss, the order of protection was highest in FM/47, intermediate in swMN/15, and lowest in goose/Qin/05 (Figure 6A). The genetic distances between the N1CC and the NA of FM/47, sw/MN/15, and goose/Qin/05 were 97.2%, 84.4%, and 82.1%, respectively. The Ad-N1CC vaccinated mice were 100% protected against death from the challenges and exhibited weight loss that was directly correlated with the genetic relationship between the N1CC vaccine and the challenge strain (Figure 6B and Table 2). These results show that vaccination with N1CC can protect against multiple highly divergent lethal HxN1 strains with multi-species origins.

## 4. Discussion

Despite the importance of influenza vaccination, a significant gap remains between optimal and current vaccine efficacy. To address this, we designed a centralized consensus NA vaccine to expand the cross-protective immune responses after vaccination. Recent studies identifying NA vaccines that lead to protective immune responses against human IAV strains [23,24,25,27,28] drove us to expand our centralized consensus platform from HA vaccines [17,18,19,20] to also design an NA-targeting vaccine. Here, we designed our NA vaccine against historical human N1 sequences. This sequence population provides a unique opportunity to test the effectiveness of a centralized consensus vaccine design against a population likely to be underrepresented in databases due to lack of sequencing technologies. By focusing on sequences prior to the emergence of pandemic H1N1 strains, we can gain insight into how this design strategy can protect against divergent N1 sequences.

We assessed antibody responses through multiple assays. Little neutralizing antibody activity after prime or boost vaccination with N1CC was detected by microneutralization. However, this assay removes the serum prior to the addition of trypsin, when cell-to-cell spread of the virus would begin. Neutralizing antibodies induced by an NA-targeting vaccine would likely not be detected through microneutralization because serum is removed prior to initiating multi-cycle infection in the assay, where NA-specific antibodies may prevent viral replication. The only neutralizing activity we observed after N1CC vaccination by microneutralization was in serum after prime vaccination against PR8. It is possible that antibody activity prevented receptor-binding through a non-specific mechanism, as has been described for anti-HA stalk antibodies preventing NA activity [42]. However, it remains unclear what this mechanism of inhibition was and why it was only observed in one animal. NA-specific antibodies induced by N1CC vaccination were assessed through ELISA using recombinant NA protein as a target. We detected robust levels of antibody present in serum that recognized NC99 NA protein, while less robust antibody concentration bound to PR8 NA was observed (Figure 3A). To investigate functional anti-NA antibody responses, we quantified NAI activity through ELLA. We detected low levels of NAI activity from our N1CC vaccinated serum against PR8 after boost, which followed a distinct trend with the sequence identity between the N1CC sequence and the strains’ NA sequences. Based on the low titers after boost, we would not expect to see any NAI activity after prime immunization alone. Unlike hemagglutination inhibition (HAI) activity, there is no current established threshold titer as a standard of protection for NAI activity [43,44]. However, NAI antibody response has previously been associated with lower disease severity in a human A/H1N1 challenge study [45]. Therefore, these antibodies induced by N1CC vaccination likely would contribute to protection from challenge in pre-clinical testing as well.

While antibody responses can prevent infection, cell-mediated immunity is crucial to limit the severity of infection. N1CC vaccination induced robust IFN-γ^+^ T-cells after a single vaccination against the genetically similar PR8 NA peptide library. No significant boosting effect was observed after a boost immunization. We did not identify cross-reactive T-cell epitopes between the HA and NA centralized consensus vaccines in this study. When compared to T-cell responses towards the PR8 HA induced by H1CC vaccination, the magnitude of NA-directed T-cell responses from N1CC vaccination were similar. Our results align with other T-cell analyses for NA vaccines [25,27,29], which show induction of antigen-specific T-cells after NA vaccination. We also observed moderate IFN-γ production in T-cells after stimulation with a divergent NA peptide pool (Viet04), suggesting that the cell-mediated immune response may contribute to protection in the absence of NAI antibody activity. More work remains to fully understand the role T-cells play in protection against influenza infection in humans [46]. However, our group and others have reported that depletion of T-cells induced by vaccination often has a negative impact on response to infection in mouse models [47,48,49]. Therefore, the presence of IFN-γ^+^ T-cells after vaccination suggests that the N1CC vaccination may provide protection from challenge even in the absence of an NAI active antibody response. Indeed, in this study, we observed low levels of NAI active antibodies after N1CC vaccination. However, with the robust IFN-γ^+^ T-cell responses observed, we saw strong protection from weight loss and mortality from genetically similar HxN1 strains and protection from mortality from divergent strains from non-human hosts. This suggests that our T-cell responses from N1CC vaccination may continue to play a key role in limiting the severity of infection in mouse models.

To further investigate protective efficacy after vaccination, we challenged vaccinated groups with a panel of HxN1 IAV strains to identify heterologous cross-protection. After a single immunization, N1CC led to complete protection from mortality against huH1N1 (PR8), swH1N1 (swUSA31), and H5N1 (Viet04) challenges. Protection from weight loss again followed a trend with the amino acid identity with the N1CC gene sequence. We expanded on this challenge panel through additional huH1N1 (FM47), swH1N1 (swMN15), and H5N1 (Goose05) strains. Consistent with the immune correlates, the more genetically related the vaccine sequence was to the challenge strain, the stronger the protection from weight loss post-infection. However, our results are notable because a single immunization with N1CC protected mice from multiple subtypes of IAV strains that originated from multiple hosts against a stringent lethal challenge of 100 MLD_50_ units. These results strongly support the further investigation of NA-targeting vaccines to induce cross-protective immunity against highly divergent strains of influenza.

Currently, development of NA-based vaccines that cross-protect against strains from multiple species is understudied. Since 2009, the human-origin H1N1 strains have all evolved from the 2009 pandemic influenza strain [6,50,51]. Therefore, understanding immunity against pdm09 strains is important for clinically relevant NA-targeting vaccines [52,53]. However, since this vaccine design did not incorporate pdm09 sequences into the design, we focused on the immune responses against strains more relevant to the design. Despite this limitation, the current panzootic outbreak of HPAIV H5N1 strains in mammals [9,54,55] shows the importance of understanding the potential for a vaccine to protect against not only relevant strains but also strains from other hosts that have the potential to circulate in humans. Therefore, despite our results only coming from older seasonal H1N1 strains, the cross-species strain immunity induced indicates high potential for future work investigating an updated N1 centralized consensus vaccine design.

In this study, we have characterized immune responses to an adenoviral-vectored, centralized consensus NA vaccine. Our results show that N1CC can induce both cross-reactive antibody and T-cell responses against HxN1 influenza strains isolated from different hosts. Further, these cross-reactive immune responses led to complete protection from mortality after challenge against a panel of lethal, divergent strains. This study represents the first evidence that a centralized consensus NA vaccine can provide protection from influenza, contributes to the recent findings that consensus NAs could be an effective strategy to improve influenza vaccine efficacy [30,31], and supports further investigation into computational vaccine designs targeting the NA for its potential to induce cross-protection against strains from multiple species. Further investigation into centralized consensus designs targeting other NA subtypes, such as N2, will allow for the delivery of a multivalent NA-targeting vaccine that could show even broader cross-protection from influenza strains of multiple subtypes and across host species.

## 5. Conclusions

Influenza vaccine efficacy is often compromised by genetic divergence between vaccine strains and those circulating during the flu season. One way to limit genetic divergence is to use a centralized consensus vaccine, which incorporates genetic diversity into the vaccine design. Previous studies from our lab that tested centralized consensus hemagglutinin vaccines found that this strategy induced broad immune responses to divergent strains. The influenza neuraminidase (NA) has also become an attractive target for vaccines due to its role in the virus replication cycle. In this study, we report the first characterization of a centralized consensus NA (N1CC) vaccine. Our results show that N1CC induces antibody and T-cell responses after vaccination and that this vaccination strategy protected mice from divergent lethal challenges from influenza viruses from multiple subtypes and from several species. These results support further investigation of centralized consensus NA vaccines for expanded sequence populations as potential protective influenza vaccines.

## Figures and Tables

**Figure 1 vaccines-13-00364-f001:**
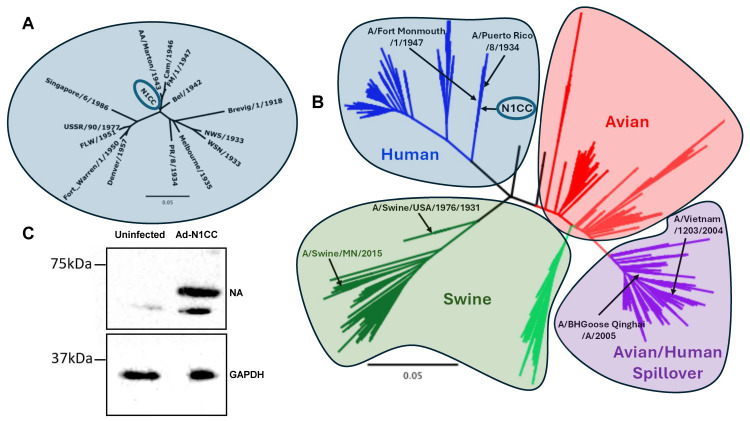
Phylogenetic analysis and protein expression from N1CC vaccine. (**A**) Neighbor-joining phylogenetic tree showing the relationship between the N1CC NA protein sequence and the representative human NA sequence population used for generation of the N1CC vaccine immunogen. (**B**) Neighbor-joining phylogenetic analysis showing unique N1 sequences isolated from humans, swine, and avian species up to 2007. Human N1 isolates (blue), swine N1 isolates (green), avian N1 isolates (red), and avian/human mixed spillover (purple) are shown. (**C**) Western blot detecting protein expression from HEK293 cells infected with N1CC. NA expression detected anti-A/New Jersey/8/1976 antiserum as a primary antibody.

**Figure 2 vaccines-13-00364-f002:**
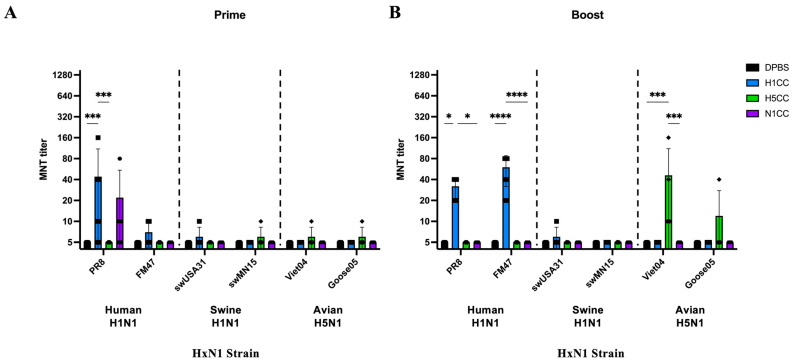
Serum neutralization of HxN1 strains measured by microneutralization. (**A**) Neutralization titers of serum against HxN1 strains after prime vaccination. (**B**) Neutralization titers of serum against HxN1 strains after boost vaccination. Pairwise comparisons were conducted using ordinary two-way ANOVA with Tukey’s correction for multiple comparisons (* *p* < 0.05, *** *p* < 0.001, **** *p* < 0.0001). Individual data points are shown. Data represented as group average plus standard deviation.

**Figure 3 vaccines-13-00364-f003:**
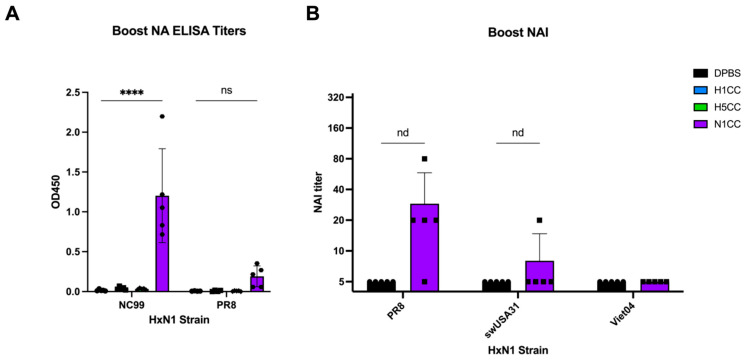
NA-specific antibodies induced after vaccination. (**A**) NA-specific antibodies measured by ELISA at single 1:100 serum dilution. ELISA plates were coated with recombinant NC99 or PR8 NA protein prior to addition of serum. (**B**) Serum inhibition of NA activity measured by ELLA. NA inhibition titers determined by the highest dilution of prime-boost serum providing 50% inhibition of NA activity against representative strains from huH1N1, swH1N1, and human-isolated H5N1 subtypes. Statistical analysis was conducted using multiple unpaired *t*-tests (nd = no discovery, ns = no significant difference, **** *p* < 0.001). Individual data points are shown. Data represented as group average plus standard deviation.

**Figure 4 vaccines-13-00364-f004:**
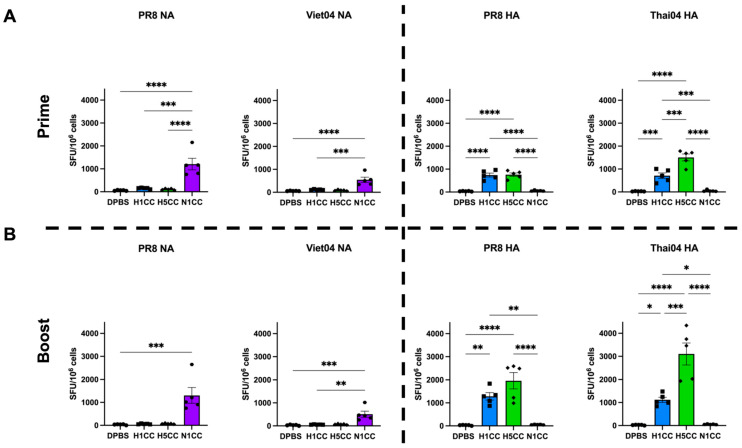
IFN-γ ELISpot analysis of T-cell response to vaccination. (**A**) Total T-cell response following re-stimulation with overlapping peptide libraries after prime vaccination. (**B**) Total T-cell response following re-stimulation with overlapping peptide libraries after boost vaccinations. Statistical analysis was conducted using ordinary one-way ANOVA with Tukey’s correction for multiple comparisons (* *p* < 0.05, ** *p* < 0.01, *** *p* < 0.001, **** *p* < 0.0001). Individual data points are shown. Data represented as group average plus standard deviation.

**Figure 5 vaccines-13-00364-f005:**
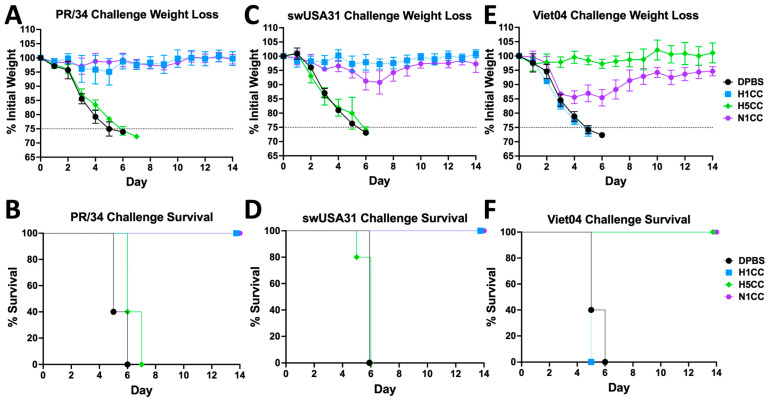
Weight loss and survival post-challenge with HxN1 influenza strains. Mice were vaccinated once with N1CC, H1CC, H5CC, or DPBS mock vaccination. Three weeks post-vaccination, the mice were challenged with PR8 (**A**,**B**), swUSA31 (**C**,**D**), or Viet04 (**E**,**F**) and monitored for 14 days post-infection. Mice experiencing ≥25% weight loss met the criteria for humane endpoint euthanasia. Weight data is group average plus or minus standard deviation.

**Figure 6 vaccines-13-00364-f006:**
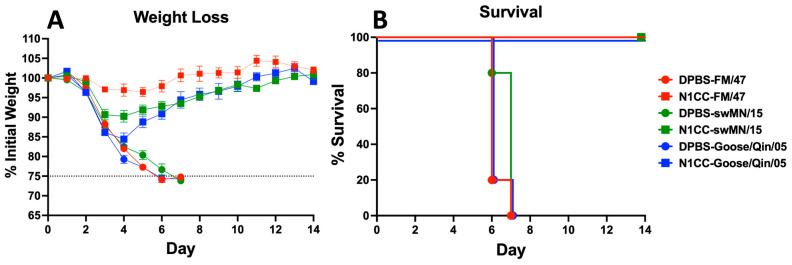
Ad-N1CC protection against HxN1 influenza virus strains. Weight loss and survival post-challenge with A/Fort Monmouth/1/1947, A/swine/USA/1976/1931, and A/bar-headed goose/Qinghai/1A/2005 influenza virus strains. Mice were vaccinated intramuscularly with 10^9^ virus particles of Ad-N1CC vaccine or DPBS. The dotted line indicates 25% baseline weight loss criteria for euthanasia. Three weeks post-vaccination, the mice were challenged with lethal influenza virus and monitored for weight loss 14 days post-infection (**A**). Mice experiencing ≥25% weight loss met the criteria for humane endpoint euthanasia (**B**). Weight data are group averages plus or minus standard deviation.

**Table 1 vaccines-13-00364-t001:** Strain names and GenBank accession IDs for the 14 H1N1 strains used as the input population for the N1CC immunogen sequence.

**Strain**	**Genbank Accession**	**Strain Name**	**Genbank Accession 2**
A/Brevig Mission/1/18	AF250356.2	A/Cam/46	CY009598.1
A/WSN/1933	J02177.1	A/Fort Monmouth/1/1947	CY009614.1
A/NWS/1933	L25815.1	A/Fort Warren/1/1950	CY009334.1
A/Puerto Rico/8/1934	J02146.1	A/FLW/1951	CY147376.1
A/Melbourne/35	CY009326.1	A/Denver/57	CY008990.1
A/Bel/1942	CY009278.1	A/USSR/90/1977	CY121880.1
A/AA/Marton/1943	CY020287.1	A/Singapore/6/1986	CY020479.1

**Table 2 vaccines-13-00364-t002:** Percentages of amino acid identities of protein sequences (HA or NA) from strains used in study to cognate vaccine immunogens (H1CC, H5CC, N1CC).

**Strain**	**Subtype**	**Abbr.**	**H1CC**	**H5CC**	**N1CC**
A/Fort Monmouth/1/1947	H1N1	FM47	95.60%	62.60%	97.20%
A/Puerto Rico/8/1934	H1N1	PR8	92.60%	63.80%	92.50%
A/swine/USA/1976/1931	H1N1	swUSA31	88.20%	64.10%	89.30%
A/swine/Minnesota/A01489606/2015	H1N1	swMN15	82.90%	62.70%	84.40%
A/Vietnam/1203/2004	H5N1	Viet04	63.10%	97.90%	82.10%
A/bar-headed goose/Qinghai/1A/2005	H5N1	Goose05	63.00%	97.70%	82.10%
A/Thailand/4(SP-528)/2004	H5N1	Thai04	63.10%	95.60%	81.90%
A/New Caledonia/20/1999&	H1N1	NC99	94.20%	62.70%	90.20%

## Data Availability

All data relevant to the study are found in the main figures or Appendix A.

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
