# Peer review of "Synthetic Neuraminidase Vaccine Induces Cross-Species and Multi-Subtype Protection"

_vaccines, 2025, doi:10.3390/vaccines13040364_

Round 1

Reviewer 1 Report

Comments and Suggestions for Authors

Comments for the authors of Vaccines manuscript vaccines-3505029:

The author of the Vaccines manuscript “Synthetic Neuraminidase Vaccine Induces Cross-Species and Multi-Subtype Protection”, present their recent work toward development of vaccines against influenza viruses.  Here, they focus their efforts on the neuraminidase surface antigen in an effort to advance centralized consensus vaccines against this virus.  This specific vaccine, known as centralized consensus neuraminidase type 1 (N1CC) is based on the ancestral N1 sequence, with delivery using human type 5 adenovirus vectors.  This N1CC vaccine elicits antibodies that inhibit NA activity and induce NA-specific T cell responses.  This vaccine protected mice against lethal H1N1 and H5N1 strains, likely through induction of immunity against the N1 expressed by these viruses.  Below are some comments that I would like the authors to address as they revise the manuscript.   

Comments:

  1. The vaccine design, vaccination scheme, and evaluation of immune responses was very good.  The only question I have is whether immunity against the pandemic H1N1 was tested.  I understand the goal of developing the N1CC against pre-2009 N1 sequences, but since this virus is readily available it would be interesting to see if immunity induced by N1CC reacts with CA09 NA.
  2. Would it be possible to test antibody isotypes (IgG1 vs IgG2a) in these vaccine recipients?  With a Th1 cytokine response reported, and low NAI antibodies against H5N1 NA, it would be interesting to know whether these antibodies are IgG2a and whether they differentially interact with NA in a way that protects without necessarily inhibiting NA activity.  This might help explain the clearance of virus seen in Figure 5E (as well as 6A for the Goose/Qin virus).

Reviewer 2 Report

Comments and Suggestions for Authors

Pekarek, et al carried out the development and evaluation of the new type of influenza vaccine by identifying centralized consensus amino acid in N1NA. This theory had been already applied in the HA with good results. Overall, under the well-designed experimental plan, the authors clearly indicate the potential of CC vaccine using the NA. These findings should be important for development of the new vaccine and vaccine strategy against seasonal and pandemic vaccine.

There are my minor comments to improve the manuscript.

L25: H1N! and H5N! should be corrected.

L178-180: Please indicate the route of boosting, intramuscularly?

L269: initial weight should be clearly described. I believe it should be weight at virus challenge.

L269-271: This sentence is not meaningful so much.

L325: Footnote of the table is not fully put onto the manuscript.

Figure 2: I prefer indicating the label of “Human (H1N1) in bottom.

Reviewer 3 Report

Comments and Suggestions for Authors

This study investigates the potential of a centralized consensus neuraminidase (NA) vaccine (N1CC) for providing cross-species and multi-subtype protection against influenza A virus (IAV). This study utilized a human adenovirus type 5 vector to deliver the N1CC vaccine and demonstrate its ability to elicit antibody and T-cell responses against multiple HxN1 influenza strains. The results showed that a single vaccination with N1CC protects mice from lethal challenge with highly genetically diverse IAV strains isolated from human, swine, and avian origins.

Some major comments are suggested to further improve the manuscript.

1、The number of samples at each time point of each experimental group is 5, which is a little small and lacks statistical significance.

2、Please supplement the controlled trial of commercial influenza vaccine.

3、Please supplement experimental data in the vaccine immune protection test to explain the virus excretion of mice at different time points. For example, by measuring the viral load in the turbinate bones and lungs to evaluate the impact of the vaccine on virus replication.

Some minor comments are suggested to further improve the manuscript.

1、There are also many formatting errors in the text, such as the lack of a period in 22 lines, which need to be checked and corrected in full.

2、There are inconsistencies in the descriptions of T cell reaction in sections 3.2 and 3.3 of the article. Please discuss them.

Comments on the Quality of English Language

The English could be improved to more clearly express the research.

Reviewer 4 Report

Comments and Suggestions for Authors

This manuscript described a way to build the influenza vaccine could induce cross-species and multi-subtype protection. The author inserts a N1CC sequence-the type 1 NA centralized consensus sequence is designed by aligned many representative types N1-to the human adenovirus type 5 vector. Although the N1CC did induce low antibodies comparable to H5CC and H1CC, N1CC could protect against HxN1 infection. I think that this design could guide us to design more effectiveness influenza vaccine.

But the following issues should be addressed:

  1. In the challenge studies, the author just used HxN1 type viruses to infect mouse, I think that adding HxN2 type may be more convinced.
  2. Author could consider to build a N2CC, and find the centralized consensus sequence between the N1CC and N2CC. This may lead a more effectiveness cross protection.
Comments on the Quality of English Language

some sentences need to be modified.

Round 2

Reviewer 3 Report

Comments and Suggestions for Authors
  1. Please supplement experimental data in the vaccine immune protection test to explain the virus excretion of mice at different time points. For example, by measuring the viral load in the turbinate bones and lungs to evaluate the impact of the vaccine on virus replication.

While we agree that the addition of this experiment would help to further explain the protection we observed, given the time constraints of the revision process, we would be unable to perform this experiment as we would need to vaccinate animals and perform the challenge studies.

If the time given by the editor is not enough, you can request an extension of the time to answer the review questions, and the new experiment results can re-validate the results of the original experiment.

Author Response

Comment 1: Please supplement experimental data in the vaccine immune protection test to explain the virus excretion of mice at different time points. For example, by measuring the viral load in the turbinate bones and lungs to evaluate the impact of the vaccine on virus replication.

Response 1: Thank you for the suggestion. Studies that examine the kinetics of replication and transmission are valuable. However, these suggested studies would go beyond the scope of the research described in the manuscript. In this study, we use a lethal influenza virus challenge model to determine the prophlyactic efficacy induced by the N1CC vaccine. Viral challenge studies, such as those included in this manuscript, are the most effective means to measure protective vaccine efficacy.